# How Curcumin Targets Inflammatory Mediators in Diabetes: Therapeutic Insights and Possible Solutions

**DOI:** 10.3390/molecules27134058

**Published:** 2022-06-24

**Authors:** Yaseen Hussain, Haroon Khan, Ghallab Alotaibi, Fazlullah Khan, Waqas Alam, Michael Aschner, Philippe Jeandet, Luciano Saso

**Affiliations:** 1College of Pharmaceutical Sciences, Soochow University, Suzhou 215123, China; pharmycc@gmail.com; 2Department of Pharmacy, Bashir Institute of Health Sciences, Islamabad 44000, Pakistan; 3Department of Pharmacy, Abdul Wali Khan University Mardan, Mardan 23200, Pakistan; waqasalamyousafzai@gmail.com; 4Department of Pharmaceutical Sciences, Al-Dawadmi Campus, College of Pharmacy, Shaqra University, Riyadh 11961, Saudi Arabia; ghalotaibi@su.edu.sa; 5Faculty of Pharmacy, Capital University of Science & Technology, Islamabad 44000, Pakistan; fazlullah.khan@cust.edu.pk; 6Department of Molecular Pharmacology, Albert Einstein College of Medicine, Bronx, NY 10461, USA; michael.aschner@einsteinmed.org; 7Research Unit “Induced Resistance and Plant Bioprotection”, Department of Biology and Biochemistry, Faculty of Sciences, University of Reims, EA 4707–USC INRAe 1488, SFR Condorcet FR CNRS 3417, P.O. Box 1039, CEDEX 02, 51687 Reims, France; philippe.jeandet@univ-reims.fr; 8Department of Physiology and Pharmacology “Vittorio Erspamer”, Sapienza University, Piazzale Aldo Moro 5, 00185 Rome, Italy; luciano.saso@uniroma1.it

**Keywords:** diabetes mellitus, curcumin, inflammatory mediators, diabetes resistance, late complications

## Abstract

Diabetes mellitus is a multifactorial chronic metabolic disorder, characterized by altered metabolism of macro-nutrients, such as fats, proteins, and carbohydrates. Diabetic retinopathy, diabetic cardiomyopathy, diabetic encephalopathy, diabetic periodontitis, and diabetic nephropathy are the prominent complications of diabetes. Inflammatory mediators are primarily responsible for these complications. Curcumin, a polyphenol derived from turmeric, is well known for its anti-oxidant, anti-inflammatory, and anti-apoptotic properties. The regulation of several signaling pathways effectively targets inflammatory mediators in diabetes. Curcumin’s anti-inflammatory and anti-oxidative activities against a wide range of molecular targets have been shown to have therapeutic potential for a variety of chronic inflammatory disorders, including diabetes. Curcumin’s biological examination has shown that it is a powerful anti-oxidant that stops cells from growing by releasing active free thiol groups at the target location. Curcumin is a powerful anti-inflammatory agent that targets inflammatory mediators in diabetes, and its resistant form leads to better therapeutic outcomes in diabetes complications. Moreover, Curcumin is an anti-oxidant and NF-B inhibitor that may be useful in treating diabetes. Curcumin has been shown to inhibit diabetes-related enzymes, such as a-glucosidase, aldose reductase and aldose reductase inhibitors. Through its anti-oxidant and anti-inflammatory effects, and its suppression of vascular endothelial development and nuclear transcription factors, curcumin has the ability to prevent, or reduce, the course of diabetic retinopathy. Curcumin improves insulin sensitivity by suppressing phosphorylation of ERK/JNK in HG-induced insulin-resistant cells and strengthening the PI3K-AKT-GSK3B signaling pathway. In the present article, we aimed to discuss the anti-inflammatory mechanisms of curcumin in diabetes regulated by various molecular signaling pathways.

## 1. Introduction

Diabetes mellitus is a chronic metabolic multifactorial disorder accompanied by altered macro-nutrient metabolism, due to the lack of insulin activity, or inability to use insulin properly [1]. The World Health Organization (WHO) indicated that about 693 million people will be diabetic at the end of 2030 [2]. Both types of diabetes mellitus (type 1 and 2) have associated complications which are life threatening and affect lifestyle [3]. These include diabetic retinopathy [4], diabetic cardiomyopathy [5], diabetic encephalopathy [6], diabetic periodontitis [7] and diabetic nephropathy [8]. Inflammation ensues when pathogens invade a living system, or tissue injury occurs, and, subsequently, infiltrate and activate the adoptive and/or innate immune system, resulting in the production of inflammatory cytokines [9]. The release of inflammatory mediators is mediated, leading to diabetic complications [10]. Thus, the pathophysiology of diabetes mellitus is linked with chronic inflammation. The role of inflammatory mediators in type 2 diabetes mellitus has been confirmed [11]. The mechanism by which inflammation leads to diabetes mellitus is not fully understood. However, certain necrotic factors, cytokines, chemokines and interleukins are involved in the mechanism of inflammation [12,13]. Inflammatory mediators, in turn, generate reactive oxygen species and oxidative stress [14]. 

Inflammation is a natural physiological reaction triggered by harmful events, such as infection or tissue damage. Inflammation is the result of a combination of factors, according to observational research. The immune system triggers a complicated set of responses. Inflammation can also lead to a variety of health problems, including both physiological and pathological morbidities. Similarly, inflammation is linked to a variety of diseases [15]. Changes in signaling pathways are linked to elevated levels of inflammatory mediators, free radicals and lipid peroxides. Inflammation is also thought to play a role in disease. In the healing of wounds and the fight against infection, inflammation plays an important role. There are two types of inflammation: acute and chronic. Acute inflammation is the first step of the inflammatory process (innate immunity), caused by the immune system’s activity, which only lasts for a short duration [16].

Curcumin is an anti-oxidant and NF-κB inhibitor that may be useful in preventing and treating diabetes. Curcumin has been shown to inhibit diabetes-related enzymes, such as a-glucosidase, aldose reductase, and aldose reductase inhibitors [17]. Curcumin has been shown to have therapeutic potential in the prevention, or delaying, of diabetic retinopathy, by acting as an anti-oxidant and an anti-inflammatory, and inhibiting vascular endothelial development and nuclear transcription factors. Curcumin’s current applications for glycemia and diabetes-related liver illnesses, neuropathy, adipocyte dysfunction, vascular diseases, nephropathy, and pancreatic disorders were described in [18]. This review article highlights the therapeutic potential of curcumin in diabetes mellitus, focusing on inflammatory mediators associated with diabetes mellitus, to provide an insight into the design of endocrine disease targeted therapy. 

## 2. Inflammation in Diabetes

Diabetes mellitus (DM) is a cluster of disorders related to carbohydrate metabolism. Defects in insulin synthesis or secretion, or in both, are key features of DM, eventually leading to hyperglycemia [19]. Diabetes mellitus is classified as type 1, type 2 and gestational diabetes. Beta cells of the pancreas secrete insulin and its specific and selective destruction is the main characterized feature of type 1 diabetes mellitus (T1DM). Type 1 diabetes is an autoimmune disease, in which pancreatic β-islet cells are destroyed by the immune system. Whilst oxidative damage might be part of the destruction mechanism, the notion that any anti-oxidant or anti-inflammatory agent could prevent, or ameliorate, the effects of the disease is disingenuous [20]. Both cellular and humoral immunity is involved in the T1DM pathogenesis. Theories on the relationship of inflammation with diabetes have focused on inflammatory infiltrates. During the pathogenesis of T1DM, insulitis (inflammation of beta cell pancreatic islets) develops [21]. In addition, both CD8+ and CD4+ T cells are involved in the pathogenesis of T1DM, targeting peptide epitopes and β-cell auto-antigens [22]. In addition, it has been demonstrated in adoptive T-cell transfer models that T1DM is associated with damage to peri-islet inflammatory infiltrates [23,24]. Dendritic and natural killer cells are also involved in the pathogenesis of T1DM, following the process of islet infiltration [25,26]. Various inflammatory mediators involved in DM are depicted in Figure 1.

Inflammation and its related pathways are involved in the pathogenesis of T2DM and the progression of diabetes is regulated by the interaction of these different cell types. During therapy, the lost parts of islets and the function of β-cells must be preserved and targeted. 

T2DM is characterized by insulin resistance, followed by insulin hypersecretion in β-cells of Langerhans. However, once diagnosed, the β-cells are unable to secrete insulin. Without a doubt, low-level inflammation is a characteristic of the disease, but it is very difficult to discern whether the association is cause or effect. For example, hyperglycemia is known to cause inflammation, but this would happen after the development of type 2 diabetes and would be, therefore, an effect of the disease. While these processes might be involved in the etiology of type 2 diabetes, the evidence is not yet clear and, therefore, the benefit of an anti-inflammatory agent in this setting is highly speculative [27]. 

Gestational diabetes mellitus (GDM) is glucose intolerance of varying degrees that appears during pregnancy. GDM is diagnosed by evaluating pregnant ladies for clinical risk markers and assessing at-risk women for impaired glucose tolerance, which is generally mild and subclinical [28]. The degree of T2DM varies concerning insulin resistance and β-cell dysfunction. Insulin resistance is the key factor in the induction of T2DM. Among other causative factors of insulin resistance, chronic inflammation is considered to be a potentially mechanistic factor, following certain inflammatory pathways like Jun N-terminal kinases (JNKs) and the transcription factor NF-kappa B (Nf-κB) [29]. In addition to the role of inflammatory cytokines in diabetes, adipokines amplify the inflammatory response and trigger inflammation-related T2DM [30]. Immune cells and macrophages induce adipose tissue infiltration that triggers inflammation, which, in turn, leads to the secretion of chemokines and cytokines, bridging a pathological link between diabetes, insulin resistance and obesity [31].

The gut microbiome plays an important role in the regulation of energy balance and immune function within the host system, thus, any alteration in gut microbiota affects immune cells and directly leads to abnormal consequences [32]. Lipopolysaccharides trigger the release of inflammatory cytokines from adipocytes and immune cells resulting in inflammation [33]. The process culminates with insulin resistance and type 2 diabetes mellitus. Loss of pancreatic β-cells plays an important role in type 2 diabetes mellitus. Interleukin-6 in pancreatic islets induces apoptosis and secretion of inflammatory cytokines triggering inflammation and, thus, serving as a key predictor in the progression of T2DM [34]. Moreover, in adipose tissues, over-production of TNF-α leads to the death of β-cells and inflammation induction that contributes to peripheral tissue insulin resistance [35]. 

Apart from this, inflammatory mediators that target certain other signaling pathways, like JNK, MAPK, p38 etc., contribute to insulin receptor substrate (IRS) degradation, through enhanced serine phosphorylation. This interrupts insulin signaling, mediates insulin resistance and, eventually, results in diabetes mellitus [36]. Here, insulin resistance is specifically attributed to inhibition of JNK1, MAPK and IKKβ activation. Tyrosine phosphorylation of Akt/GSK-3 is inhibited by angiotensin II-mediated inflammation that results in L6 myotube insulin resistance [37]. In summary, all these findings reveal that insulin resistance and inflammation have an important and consistent connection with each other that leads to diabetes mellitus.

In a research study it was found that pancreatic beta cell neogenesis was significantly interrupted by free radicals, indicating the role of free radicals in the proliferation of pancreatic beta cells [38]. Findings of a recent research study suggested that oxidative stress damaged beta-pancreatic cells. In vitro activities indicated the disturbance in beta cells through oxidative stress. In addition, hydrogen peroxide in a concentration of 200 µM reduced insulin secretion stimulated by glucose [39]. In a nutshell, in diabetic patients, new therapeutic intervention could be targeted such that beta cell dysfunction, mediated by inflammation, is considered to be a tuning site. It is suggested that the use of an agent that controls inflammation and inflammatory mediators at the pancreatic site would, hopefully, induce glucose homeostasis and, eventually, improve the quality of life of diabetic patients. 

## 3. Therapeutic Potential of Curcumin: An Overview

Curcumin is a lipophilic polyphenol and secondary metabolite extracted from *Curcuma longa* (turmeric). It imparts the yellow color to turmeric and is present in abundance, along with other curcuminoids, making up 10 to 20% of curcumin used commercially. Semi synthetic derivatives of curcumin are also available, that are categorized as nutraceuticals [40]. The chemical structure of curcumin is shown in Figure 2. 

Curcumin is effective against inflammation and edema. In a research study, vasodilatation was induced in mice ears using dimethylbenzene. The resultant effect was treated with curcumin in a concentration of 0.05–1 g/kg dose for 7 days [41]. Curcumin is acceptable at large doses (12 g/day) in humans, according to phase I clinical studies; however, its bioavailability is limited [42]. Inadequate absorption, fast metabolism, and rapid systemic clearance seem to be significant factors contributing to low curcumin concentrations in the plasma and tissues. Curcumin’s bioavailability has been improved by the use of the following: curcumin nanoparticles, structural analogues of curcumin (e.g., EF-24), curcumin phospholipid complexes, adjuvants, such as piperine, to prevent glucuronidation, and, lastly, liposomal curcumin. Curcumin structural analogues (e.g., EF-24) have been shown to have a fast absorption rate and a short plasma half-life. Notwithstanding its poor bioavailability, curcumin has been shown to have medicinal potential against a variety of human ailments, particularly diabetes, cancer, arthritis, cardiovascular disease, Crohn’s disease, and neurological disease. Curcumin’s increased bioavailability is likely to propel this potential natural substance to the frontline of curative medicines for human illness therapy in the near future [42].

Curcumin was evaluated for its anti-inflammatory potential in a rat model for the treatment of osteoarthritis. Results suggested that curcumin significantly reduced the expression of cytokine levels in synovial fluid targeting the TLR4/NF-κB signaling pathway [43]. The effect of curcumin on inflammatory indices was evaluated in a randomized control study. Results of the study indicated an outstanding reduction in inflammation through reduction in TNF-α, concluding that curcumin plays a key role in inflammation suppression in hepatic patients with nonalcoholic fatty liver disorders [44]. Systemic infection, such as sepsis, is a crucial factor in organ failure, such as sepsis acute kidney injury (SAKI). The therapeutic effect of curcumin was explored in SAKI using mice and SAKI cell models. The findings of the study revealed that curcumin reduced the expression of IL-6, TNF-α, and the NF-κB signaling pathway and reduced the rate of cell apoptosis resulting in the healing of injured kidney cells [45]. Apart from individual therapy, curcumin in combination therapy has shown significant anti-inflammatory action. In this regard, hyperlipidemia-induced inflammation was targeted by the combined delivery of curcumin and rutin in Wistar rats. Results of the study showed an increase in HDL and a decrease in triglyceride level after treatment with curcumin and rutin combined therapy [46]. It is concluded that curcumin has potential in treating inflammation and can be used as a therapeutic medicament.

The therapeutic effects of curcumin on cerebral ischemia/reperfusion (I/R) injury, associated with water content and infarction volume, was investigated in a study that evaluated the expression of NF-κB and Nrf2. It was concluded that the protective effects of curcumin against cerebral I/R injury imitate various properties, such as anti-oxidant potential, anti-inflammatory activity and anti-apoptotic activity, by elevating Nrf2 expression and down-regulating NF-κB [47]. In another study, the protective role of curcumin was investigated against traumatic brain injury (TBI), by targeting the p38/MAPK signaling pathway. It was found that curcumin significantly ameliorated secondary brain injury after TBI, including its effect on the neurological severity score and inflammation. Similarly, it was revealed that curcumin showed anti-inflammatory potential which was evidenced by the inhibition of inflammatory factors, such as interleukin-1β, IL-6 and TNF-α. Furthermore, curcumin remarkably decreased the expression of p-p38, based on the data collected from western blotting and immunohistochemical analysis. It was concluded that curcumin significantly improved post-traumatic inflammation and, hence, showed therapeutic potential for treating the inflammation associated with TBI [48].

Asthma and allergies are mainly mediated through inflammatory cytokines. Curcumin has been reported as having regulatory potential against these inflammatory cytokines in allergy and asthma. The respective mechanism involved is interleukin inhibition and regulation of histamine release from mast cells [49]. A psoriatic skin lesion mice model under progranulin deficient condition was used in a recent research study. In the absence of progranulin, an increased level of pro-inflammatory cytokines, epidermal thickening and a change in cell differentiation was observed. Curcumin interestingly alleviated progranulin deficiency-induced exacerbation of psoriasis [50]. This means that curcumin has a protecting role in lesion exacerbation during psoriasis. Similarly, curcumin was evaluated for its topical delivery in psoriasis. A nanoemulgel of curcumin was fabricated, via the low-energy emulsification method, and integrated into hydrogel. The results of the study showed improved efficacy of curcumin in psoriasis from the nano platform [51]. Another research study on cutaneous infections was carried out that suggested the important role of curcumin as a therapeutic agent. Curcumin effectively worked as an adjuvant therapeutic during combinatorial delivery with imiquimod. Curcumin improved the delivery of imiquimod, and reduced the worst side effects associated with imiquimod when it is delivered alone [52]. In a combinatorial delivery system, curcumin–caffeine combination, in the form of a nano sponge-based topical gel, was explored for anti-psoriatic activity using an imiquimod—induced psoriasis mouse model. The results indicated a significant reduction in the induction time for anti-psoriatic activity, from 20 to 10 days [53]. 

During the pathogenesis of cancer, multiple signaling pathways are involved and curcumin represents a potential candidate for the regulation of these signaling pathways. Among these, pro-inflammatory transcription factor (NF-κB) is involved in breast cancer cell proliferation. Curcumin down-regulates the NF-κB signaling pathway, thus, affecting the cell proliferation and invasion contributing to breast cancer treatment [54]. In another breast cancer model, curcumin induced autophagy through down-regulation of Akt protein, posing a significant management strategy for breast cancer [55]. These findings suggest the therapeutic potential of curcumin following multiple signaling pathways. The therapeutic effect of curcumin has also been exploited in lung cancer. A mechanistic approach was used to study curcumin’s anti-cancer potential, by targeting JAK2/STAT3 and NF-κB signaling pathways, in the A549 lung cancer cell line [56,57]. In addition, curcumin, via PI3K/Akt signaling suppression and microRNA-192-5p up-regulation induced apoptosis in non-small cell lung cancer cells with inhibition of cell proliferation [58]. Studies showed that elastases, such as neutrophils and α1-antitrypsin elastases, are actively involved in the proliferation of the lung tumor-approaching inflammation mechanism [59]. In vitro and in vivo studies showed that curcumin, via up-regulation of α1-antitrypsin elastases and repression of neutrophil elastases coped with lung cancer [60]. A recent study directly sprayed curcumin liposomal dry powder and gemcitabine into a rat with lung cancer. The liposomal dry powder of curcumin showed an enhanced anti-cancer activity, as compared to conventional gemcitabine. The experimental targets were TNF-α, Bcl-2 and Caspase-3 [61]. Curcumin targeting of other cancers is shown in Table 1.

### 3.1. Curcumin in Diabetes

Curcumin is a natural lipophilic polyphenol, extracted from turmeric, exhibiting many therapeutic activities, such as anti-cancer [68], anti-ulcer [69], anti-microbial [70], anti-diabetic [71] and anti-oxidant activity [72]. Results from diabetic patients and cultured in vitro cells have shown the protective effect of curcumin in diabetes mellitus [73]. In the context of diabetes prevention, curcumin was evaluated in several diabetic patients, and results showed a reduction in diabetic patient numbers after nine months of intervention of curcumin at the pre-diabetic stage [74]. It reduces inflammatory mediators and factors, like TNF–α, INF-γ, systemic inflammatory biomarkers, such as RANTES, and other cytokine levels [75]. 

Curcumin has displayed a versatile role in many ailments and, as discussed above, exhibits anti-inflammatory, anti-apoptotic and anti-oxidant potential [76]. Pre-clinical and clinical studies have shown its therapeutic potential against diabetes and its related complications [77]. Regarding diabetes, curcumin focuses on certain primary mechanisms, which are discussed below in detail. Curcumin is involved in the improvement of pancreatic beta cell function through targeting inflammatory mediators. This influence at the cellular level is depicted in Figure 3. Both oxidative stress and inflammatory mediators, which are interlinked in diabetes mellitus, are influenced by curcumin. 

Curcumin also normalizes inflammatory cytokines like IL-6 and IL-1β. In addition, curcumin targets inflammatory mediators in insulin resistance. Curcumin restores dysfunction associated with Nrf2, that promotes diabetes resistance management [78]. Curcumin has shown therapeutic potential in the management of resistant diabetes mellitus, targeting inflammatory mediators. Curcumin inhibits inflammatory signaling and the expression of Kelch-like ECH-associated protein 1 (Keap1), resulting in activation of the Nrf2 system. The latter shows effects on insulin sensitivity and redox homeostasis, through inhibition of inflammatory signaling via up-regulation of Keap-1 and activation of Nrf2 [79]. 

The therapeutic potential of curcumin nanoemulsion (NC) has been investigated in a study by administering it via the oral route into middle cerebral artery occlusion and reperfusion (MCAO/Re)-induced cerebral damage in rats caused by streptozotocin-induced diabetes. The results of the study revealed that oral feeding of MCAO/Re diabetic rats with NC was linked with significant reduction of the neurological deficit score along with brain imbalance of redox homeostasis. It was also shown that NC treatment was associated with a down-regulation of TNF-α, IL-1 β, COX-2, cleaved caspase-3 and NF-κB. Similarly, the expression pattern of glucose transporter-1protein was reinstated upon treatment [80]. 

It was found in a study that curcumin supplements changed the hepatic gene expression profiling, particularly the insulin signaling and senescence pathways. In the same study, how curcumin works to affirm the sensitivity of insulin was investigated. It was concluded that curcumin supplements up-regulated the expression of hepatic insulin-degrading enzyme (IDE) and conserved islet integrity. It was suggested, based on the findings of the study, that the diverse therapeutic potential of curcumin can be applied as a protective tool in the management of age-induced metabolic diseases [81]. 

Activation of tissue necrosis factor leads to induction of oxidative stress and inflammation, while excessive fatty acid leads to induction of inflammation. Curcumin acts on both free fatty acid and tissue necrosis factors, leading to inhibition of inflammation and inhibition of the oxidative stress state. Abbreviations: TNF-α: tumor necrosis factor-α; NF-κB: nuclear factor kappa-light-chain-enhancer of activated B cells; IRS1: insulin receptor substrate 1; AKT: protein kinase B.

### 3.2. Influence of Curcumin on Inflammatory Mediators

Inflammatory mediators are actively involved in diabetes and its associated complications. To explore the anti-diabetic activity of curcumin in its targeting of inflammatory mediators, researchers carried out experimental work on diabetic rats with periodontitis. The study was conducted on one hundred rats and gingival tissues were examined for the quantification of inflammatory mediators after curcumin treatment. Results of the study indicated the lowest concentration of TNF-α and INF-γ mediators [82]. Curcumin modulated inflammatory mediators in diabetes mellitus-related periodontitis. In another study on diabetes mellitus, comparative research was carried out on curcumin solution and liposomal curcumin formulation administered to streptozotocin-induced diabetic rats. The curcumin content was 1 mg/0.1 kg and 2 mg/0.1 kg of body weight for curcumin solution and liposomal formulation, respectively. Significant reduction in the level of cytokines, TNF-α and systemic inflammation biomarkers, such as RANTES, was noted [83]. 

Diabetic retinopathy is a common complication of diabetes mellitus that causes visual disturbances. The actions of analogs of curcumin were explored for their normalization of inflammatory mediators in a mouse model and cell culture studies. It was found that curcumin analogs significantly normalized inflammatory mediators, such as TNF-α and IL-6 in diabetic animals, and offered this phytochemical as an alternative therapy for diabetes mellitus and its associated complications [84]. 

Curcumin-loaded chitosan nano particles effectively prevented inhibition of tissue remodeling through reduction in the level of reactive oxygen species, via controlling inflammatory responses in [85]. Results suggested that curcumin could be used as a potential candidate in the treatment of type 2 diabetes mellitus and its related wound complications, targeting inflammatory mediators. Similarly, curcumin-loaded chitosan nano particles were investigated in a streptozotocin-induced rat diabetic model targeting inflammatory mediators. The fabricated curcumin nano particles significantly reduced diabetes-related wounds through a reduction in macrophage-induced inflammation [86]. Hyperglycemia in diabetes induces inflammation through multiple mediators that lead to diabetic cardiomyopathy and nephropathy. J17, a structurally similar analog of curcumin, was evaluated in streptozotocin-induced diabetic mice for its anti-inflammatory mediator activity. Results indicated that, by targeting Akt and p38 signaling pathways, J17 remarkably reduced inflammatory cytokines and other renal and cardiac dysfunction-related biomarkers that led to an acknowledgment of the anti-diabetic activity of this curcumin-related compound [87].

Combination therapies of curcumin with other anti-diabetic agents have shown a remarkable effect in focusing on inflammatory mediators. Cao et al. administered a single dose of curcumin and metformin 100 mg/kg of body weight using the oral gavage route of administration in gentamicin-induced diabetic Wistar rats. The combination therapy of curcumin and metformin improved renal function, via reduction in inflammatory mediators, such as IL-6, TNF-α and IL-1β [88]. To evaluate the role of combinatorial curcumin in regulation of inflammatory mediators in diabetic retinopathy, 28 vitreous biopsies were collected from diabetic patients and curcumin, along with homotaurine and Vitamin D3, were administered in an incremental dose of 0.5–1 µM concentration. This combinatorial therapy of curcumin showed anti-inflammatory potential against various mediators, such as interleukin-2, interleukin-6 and TNF-α, as compared to individual administration [89]. Karolina and her co researchers performed a meta-analysis on the anti-oxidant potential of curcumin in clinical trials. They found that natural curcumin does have the ability to lower MDA levels while also increasing anti-oxidant potential. [90]. In a study, curcumin treatment prevented neuronal cell death in cognitive deficit diabetic rats via modulation of several mediators [91] and, therefore, suggested its potential role in diabetic neuro-inflammation. 

Evidence suggests that low-grade chronic inflammation caused by adipose tissue cytokine release plays a key role in exacerbating the inflammatory aspects of obesity-related cardiovascular disease and insulin resistance [92]. Obesity has been linked to an increase in serum levels of the pro-inflammatory mediator prostaglandin E2 [93]. In obese mice models, knocking down pro-inflammatory genes slowed the onset of insulin resistance and hyperglycemia [94]. Curcumin contains anti-inflammatory properties and can be used to treat inflammatory illnesses. It has been demonstrated to block the activation of the c-Jun N-terminal kinase (JNK) pathway as well as the expression of NFB p65, resulting in a reduction in pro-inflammatory responses [95]. It also inhibits macrophage activation and fat tissue invasion. Curcumin prevents NFB translocation to the nucleus by decreasing the phosphorylation of protein IB kinase (IKK) [92]. Curcumin administration resulted in significant reductions in TNF, interleukin-1 (IL-1), IL-6, and cyclooxygenase-2 gene expression. 21 Reduced IL-1 expression is especially important, because it ensures insulin receptor substrate-1 (IRS-1) functionality in adipocytes [92].

Curcumin and other turmeric extracts have been suggested to be powerful PPAR agonists that mediate adipocyte differentiation. Curcumin, however, appears to be unable to boost PPAR or cause pre-adipocyte differentiation, according to new research [96]. Curcumin activates AMPK before inhibiting PPAR-***γ***, according to a recent study. AMPK inhibits the activation of adipogenesis-related transcription factors like PPAR and CCAAT/enhancer binding protein a (C/EBP) [97]. In 3T3-L1 adipocytes, activated AMPK, an upstream regulator of PPAR-***γ***, was discovered to be an inhibitor of adipocyte development. Sincse AMPK is involved in maintaining cellular energy balance and adipocyte death, curcumin-induced AMPK-mediated suppression of adipocyte proliferation has clinical implications in the treatment of obesity [98]. 

A graphical illustration of curcumin influencing inflammatory mediators in diabetes is shown in Figure 4.

### 3.3. Role of Curcumin Targeting Inflammation in Diabetes Resistance

Insulin resistance has a prominent role in diabetes resistance, and was treated by curcumin through a reduction in the expression of SOCS3, in STAT3 signaling, by increasing the level of IRS-1 and Rac-1 and by suppression of phosphorylation of the ERK/JNK signaling pathway [99,100]. In a streptozotocin and high fat diet rat model, type 2 diabetes was induced. The genes and protein expression patterns were evaluated through real-time PCR and western blot analysis. Curcumin administration reduced SOCS3 and STAT3 expression and elevated IRS-1, Rac1, and Rac-GTP levels, representing its anti-diabetic activity. The increased level of IRS-1, due to curcumin, was found to be responsible for the alleviation of insulin resistance. In addition, the enhanced level of Rac1, due to curcumin, restored normal glucose tolerance and insulin sensitivity [101]. To improve insulin sensitivity, curcumin and its metabolites were explored for resistance to diabetes using HepG2 cells. Curcumin and its related metabolites improved insulin sensitivity by suppressing phosphorylation of ERK/JNK in HG-induced insulin-resistant cells and strengthening the PI3K-AKT-GSK3B signaling pathway [102]. 

Insulin resistance is usually caused due to impairment in the signaling pathways related to insulin so that the target tissues pose lesser, or no, response to circulating insulin. Diabetes mellitus and insulin resistance are concerned with aging, obesity and inflammation [103,104]. Various insulin resistance in vitro models have shown that Nrf2 dysfunction results in insulin resistance, thus, exhibiting a key role in diabetes resistance [105,106]. In this context, curcumin, being a natural and cheap source, was evaluated for its role in diabetes resistance in animals. Results of the study indicated that curcumin inhibits inflammatory signaling and the expression of Kelch-like ECH-associated protein 1 (Keap1), resulting in activation of the Nrf2 system. Insulin sensitivity and redox homeostasis were significantly affected by inhibition of inflammatory signaling and the mediated up-regulation of Keap-1 induced by curcumin through activation of Nrf2 [107]. Curcumin intervention significantly restored the anti-oxidative activity of Nrf2.

Eighty overweight subjects with suboptimal fasting plasma glucose were evaluated for curcumin effect in controlling insulin resistance using a randomized double-blind placebo-controlled trial. A 200 mg phytosomal curcumin tablet was administered to subjects for eight weeks. Results indicated an improved hepatic glycemia function in impaired fasting glucose subjects with insulin sensitivity. The insulin sensitizing effect was attributed to the anti-inflammatory activity of curcumin [108].

Curcumin and prebiotic *Lactobacillus acidophilus* were co-delivered for the treatment of insulin resistant experimental animals with diabetes. Metabolic syndrome was induced through the administration of 20% fructose along with water and the animals were treated with curcumin (100 mg/kg/day) and *Lactobacillus acidophilus* (2 × 10^8^ cfu/mL/day) individually, as well as in combination for the last four weeks. Apelin, resistin and insulin were determined by ELISA test kits, while glucose, triglyceride and total protein were determined by auto analyzer. The results showed that co-delivery of both curcumin and *Lactobacillus acidophilus* improved hormone levels and minimized the insulin resistance score in the rats [109]. Curcumin loaded nanomicelles were targeted to insulin resistant male Wister rats. Curcumin from a nano platform showed significant outcomes (improved insulin resistance) in the treatment of insulin resistance and diabetes mellitus [110]. A recent study on phytochemicals has shown that curcumin is actively involved in decreasing insulin resistance, HbA1c and fasting insulin, leading to effective diabetes management [111]. T2DM is a heterogeneous metabolic disorder, characterized by insulin tolerance and cell dysfunction, resulting in persistent hyperglycemia. Hyperglycemia is thought to be the primary cause of metabolic disorders, which not only reduce living standard and longevity, but also place a financial strain on health-care systems [112]. 

Curcumin has been shown in previous research to be a safe phytochemical that can help with obesity-related insulin resistance, inflammation, and oxidative stress. The goal of this study was to see if curcumin might prevent BPA-induced insulin resistance in vitro and what the mechanism was. The protective benefits of curcumin were determined when in vitro insulin resistance was established with BPA treatment in human liver HepG2 cells. We found that treating HepG2 cells with 100 nM BPA for 5 days reduced glucose consumption, impaired insulin signaling, increased pro-inflammatory cytokines and oxidative stress, and activated signaling pathways. Inhibiting the JNK and p38 pathways, but not the ERK or NF-B pathways, improved glucose consumption and insulin signaling in BPA-treated HepG2 cells. Curcumin was found to effectively reduce the range of consequences of BPA-induced insulin resistance, although pre-treatment with the JNK and p38 agonist, anisomycin, could greatly compensate for curcumin’s effects. These findings demonstrated the importance of JNK/p38 activation in BPA-induced insulin resistance and pointed to curcumin as a possible candidate for BPA-induced insulin resistance treatments [113]. A brief summary of curcumin effects on diabetic complications is shown in Table 2.

### 3.4. Nano-Curcumin Targeting Inflammatory Mediators 

Nano-curcumin is a promising therapeutic candidate having anti-inflammatory, anti-cancer, anti-amyloid, anti-oxidant, anti-bacterial, and anti-fibrosis capabilities, and it has the potential to prevent and treat a wide range of human disorders [120,121,122,123,124]. Curcumin is a possible anti-inflammatory drug, with anti-inflammatory effects mediated via inhibition of enzyme activity, cytokine synthesis, and transcription factor activation [125]. In a study, curcumin-solid lipid nanoparticles (CSLNs) were synthesized and their efficacy was improved in an allergic model of asthma-induced ovalbumin. The experimental results revealed that CSLNs suppressed the airway hyper-responsiveness and inflammatory cell infiltration. Similarly, it was revealed that CSLNs primarily blocked the expression of T-helper-2-type cytokines (interleukin-4 and 13) in bronchoalveolar lavage fluid [126]. In another study, it was confirmed that nano-curcumin is effective against esophageal adenocarcinoma (EAC) cell lines, OE33 and OE19. Such nano-curcumin sensitizes EAC cells to T cell-induced cytotoxicity and reduces the pro-inflammatory signals originating from T cells [127]. 

In RAW 264.7 murine macrophages, CUR-SLNs had better solubility than their natural form and dramatically reduced LPS-induced pro-inflammatory mediators (NO, PGE2, and IL-6) through inhibiting NF-kB activation [128]. Similarly, in another study, it was revealed that nano-curcumin improves oral bioavailability and, hence, efficiency in the prevention of streptozotocin (ST)-induced diabetes in rats, at least in part, by suppressing inflammation and pancreatic beta-cell death, as compared to the native form. Loss of NF-kb activation causes down-regulation of COX-2 and iNOS expression, blocking the inflammatory response and carcinogenesis [129]. Curcumin-loaded PLGA nanoparticles (CUR-NPs) were shown to reduce pro-inflammatory mediators in staphylococcus aureus-affected mammary tissues, via boosting NF-kb signaling, in experimental research [130]. 

In addition, CURNP appears to be an effective alternative for native curcumin in the treatment of murine mastitis. Furthermore, it was found in a study that curcumin encapsulated in nano micelles exhibited higher anti-inflammation action than curcumin alone in preventing paraquat (PQ)-induced lung damage [131]. Similarly, another study gave evidence that the anti-inflammatory action of curcumin liposomal formulations (CurLIPs) in human dental pulp stem cells, in response to 2-hydroxyethyl methacrylate (HEMA) treatment, increased the quality of dental care, with a significant human community benefit [132].

In a study, curcumin was developed and characterized as a self-nano emulsifying drug delivery system (SNEDDS) to improve its bioavailability, and then tested for efficacy in experimental diabetic neuropathy. Male Sprague Dawley rats were used in the bioavailability trials. In addition, different indicators, such as nerve function and sensory perception, were measured, as well as inflammatory proteins (NF-B, IKK, COX-2, iNOS, TNF, and IL-6) to determine the efficiency of the formulation in diabetic neuropathy. Longer plasma exposure and bioavailability were achieved using a nanotechnology-based formulation. When compared to naive curcumin, SNEDDS curcumin produced greater benefits against functional, behavioral, and biochemical deficits in experimental diabetic neuropathy. The increased neuroprotective effect of SNEDDS curcumin was validated by western blot analysis. Due to its greater bioavailability, the SNEDDS curcumin formulation was demonstrated to provide better protection against diabetic neuropathy [133].

In vivo wound healing tests in streptozotocin-induced diabetic mice revealed quick wound healing efficacy when compared to conventional formulations, in another investigation. Histopathological investigations also demonstrated its potential to restore normal skin structure and tissue histological conditions. As it combines the anti-oxidant and anti-inflammatory effects of curcumin, the silk fibroin-based nanofiber wound dressing appears to be an appropriate preparation in combination with curcumin. As a result, the silk fibroin-based nanofiber loaded with curcumin was found to have a high healing capacity in diabetic wounds [134]. 

Due to its ease of surface modification, ability to blend with a variety of polymers, non-immunogenicity and non-toxicity, and substantial compatibility with cells and tissues, chitosan is most often utilized for the delivery of anti-diabetic phytocompounds. Chitosan nanoparticles (NPs) encapsulating ferulic acid or curcumin were prepared using a bottom-up ionic gelation technique. According to Chauhan et al., curcumin chitosan NPs were found to have a superior effect on the translocation of GLUT4 to the cell surface in L6 rat skeletal muscle cells in vitro [135].

### 3.5. Clinical Studies 

Curcumin has shown outstanding therapeutic potential in various stages of late diabetes complications [136]. In addition to this, in a clinical study in Iran, curcumin treatment of 40 women with type 2 diabetes, for eight weeks, caused a significant reduction in blood glucose levels [137]. However, the study needs further detailed exploration on a large scale with effects in both sexes. Similarly, a double-blind three-month clinical study of curcumin regarding 50 patients with metabolic syndrome caused significant modulation of inflammatory mediators [138]. 

## 4. Conclusions and Future Perspectives 

Inflammatory mediators are involved in the induction of diabetes mellitus and such consequences as diabetic retinopathy, diabetic cardiomyopathy, diabetic nephropathy, diabetic encephalopathy, diabetic periodontitis and endothelial dysfunction. Curcumin has shown efficient anti-inflammatory, anti-apoptotic and anti-oxidant potential. In patients with metabolic syndrome, short-term treatment with a curcuminoid-piperine combination reduced inflammatory status dramatically. Curcuminoids are effective, safe and natural CRP-lowering compounds. In addition, curcumin reduces levels of inflammatory mediators and factors, such as TNF-α, INF-γ, reduces systemic inflammatory biomarkers. such as RANTES. and other cytokine levels. Curcumin also reduces the levels of inflammatory cytokines, such as IL-6 and IL-1β, and targets inflammatory mediators in insulin diabetes resistance. Curcumin restores dysfunction associated with Nrf2 that helps in diabetes resistance management. Similarly, curcumin inhibits inflammatory signaling and the expression of Kelch-like ECH-associated protein 1 (Keap1), resulting in activation of the Nrf2 system. This affects insulin sensitivity and redox homeostasis through inhibition of inflammatory signaling, up-regulation of Keap-1 and activation of Nrf2.

Curcumin metabolites have shown significant results in diabetes mellitus treatment focusing on inflammatory mediators. Thus, there is a need for future investigations to explore the structural activity relationship of curcumin to develop new curcumin agents in this context that are more effective than natural curcumin. Curcumin, due to its low bioavailability, has been evaluated from a nano platform and very few studies are available in targeting inflammatory mediator-induced diabetes mellitus. Future studies are required in this regard, as are their respective clinical trials. Resistant diabetes has not been sufficiently explored and evaluated in the targeting of inflammatory mediators because determining anti-inflammatory activity is not sufficient and valid for such a daunting dilemma. In a nutshell, further explorations regarding the issues discussed could open new avenues for curcumin delivery in diabetes mellitus and help researchers find new approaches to lessen the burden of diabetes-related complications.

## Figures and Tables

**Figure 1 molecules-27-04058-f001:**
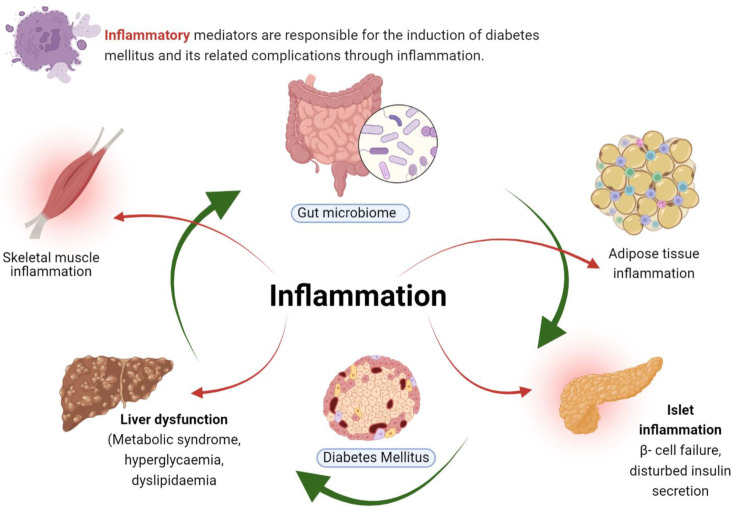
Diabetes mellitus and its inflammatory mediators. A variety of inflammatory cytokines are involved in the death of pancreatic β-cells that are activated via immune cells. In such immune disorder, the regulatory T-cells pose a defect while the effector T-cells target multiple β-cell auto-antigens and peptide epitopes that contribute to T1DM development. In addition, during the progression of T1DM, the immune β-cell profile also alters and macrophages become critical mediators of inflammation in islet cells because of reactive oxygen species generation in β-cells. This action on β-cells is considered to be a toxic one. Apart from this, dendritic cells and natural killer cells also contribute actively to this process.

**Figure 2 molecules-27-04058-f002:**
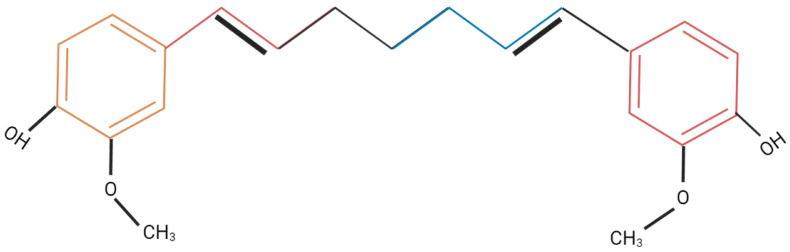
Chemical structure of curcumin.

**Figure 3 molecules-27-04058-f003:**
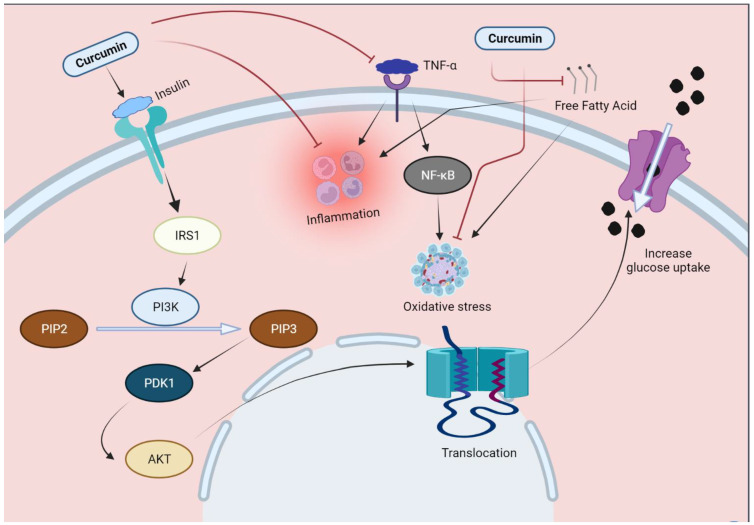
Curcumin effect on inflammatory mediators, and its cellular effects.

**Figure 4 molecules-27-04058-f004:**
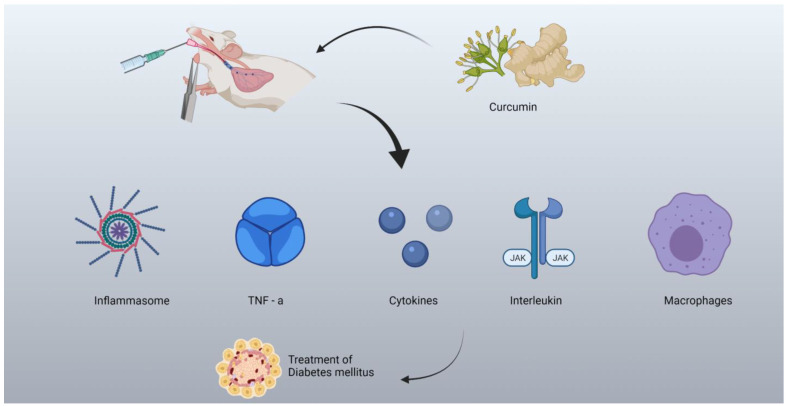
Curcumin targeting various inflammatory mediators in diabetes mellitus.

**Table 1 molecules-27-04058-t001:** Therapeutic potential of curcumin in various cancers.

Cancer Type	Mechanistic Pathway	Effects	References
Myelogenous leukemia	modulation of the PTEN/AKT via mediation of miR-21	Tumor suppression	[62]
Gastric cancer	Bax up-regulation, Bcl-2 down-regulation, PAK1 inhibition, cell cycle arrest at G2/M phase	Cell apoptosis, proliferation suppression and invasion	[63,64]
Colorectal cancer	Bcl-2, ROS, JNK	Apoptosis survival reduction	[65]
Hepatic cancer	MAPK, ERK1/2, ERK5	Regulation of epithelial to mesenchymal transition	[66]
Brain tumor	PI3K/Akt, NF-κB, Bcl-xL	Mitochondrial dysfunction	[67]

**Table 2 molecules-27-04058-t002:** Summary of the studies of the effects of curcumin on diabetic complications.

STZ-Induced Diabetic Nephropathy in Rat			
Dose Used	Assays	Effects	Reference
Curcumin (100 mg/kg and Gliclazide (10 mg/kg, orally)	Tail flick and tail pinch tests, Hot plate, peroxynitrite, C-peptide, TNF-α levels, lipid peroxides.	Improvement in sensory motor deficits, increased c-peptide levels and decreased peroxynitrite, lipid peroxides and TNF-α levels	[114]
Apocyanin (2.5 mg/kg), i.p and Curcumin intragastric administration (200 mg/kg)	Expression of NADPH oxidase, Hydrogen peroxide, Paw withdrawal threshold, SOD levels and MDA in spinal cord.	Improved expression of p47phox and gp91phox of NADPH oxidase. Paw withdrawal threshold improved. MDA, H_2_O_2_ levels decreased and SOD levels augmented	[115]
Curcumin (60 mg/kg; p.o.)	NO and TNF-α in brain homogenate, Tail immersion and hot plate assay	Reduce in nitrite and TNF-α levels, decline in tail withdrawal reflex and paw licking.	[116]
Curcumin (15–30 mg/kg, p.o. for 2 weeks)	BUN, Creatinine, urinary albumin excretion, renal malondialdehyde urea clearance, SOD, catalase & glutathione levels.	Improved urea and creatinine clearance. Augmented antioxidant enzymes and abridged MDA level	[117]
Curcumin (100 mg/kg/day, p.o. for 8 weeks)	p-AMPKα (Thr 172), Expression of SREBP-1c, AMPKα, vascular endothelial growth factor (VEGF) and adipose differentiation-related protein (ADRP) and Immunofluorescence for type IV collagen and FN	Prohibited expression of SREBP-1c and decreased expression of acetyl CoA carboxylase, fatty acid synthase and ADRP Amplified phosphorylation of AMPK, suppressed expression of ECM, VEGF and TGF β proteins, such as type IV collagen and FN	[118]
Curcumin (50 mg/kg/day for 6 weeks)	BUN, Albumin and creatinine, Superoxide dismutase, lipid peroxide levels. Western blot analysis of HSP-27, H3, p38	Reduced BUN and creatinine and improved albumin. Decreased MDA and enhanced SOD level. Reduced p38 and HSP-27 expression, decrease in dephosphorylation and increased acetylation of histone H3	[119]

BUN: Blood urea nitrogen, MCP-1: Monocyte chemo attractant protein-1, TGF-β1: Transforming growth factor-β1, VEGF: Vascular endothelial growth factor, FN: Fibronectin, HSP: Heat shock protein. ERK: Extracellular related kinases, PKC: Protein kinase c, NAG: N-acetyl glucosamine, COX-2: Cyclooxygenase-2, IL-6: Interleukin-6, NF-κB: Nuclear factor kappa, NADPH Oxidase: Nicotinamide adenine dinucleotide phosphateoxidase, TNF-α: Tumor necrosis factor-α.

## Data Availability

Not applicable.

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
