# Peer review of "How Curcumin Targets Inflammatory Mediators in Diabetes: Therapeutic Insights and Possible Solutions"

_molecules, 2022, doi:10.3390/molecules27134058_

Round 1

Reviewer 1 Report

The manuscript entitled "How Curcumin Targets Inflammatory Mediators in Diabetes: Therapeutic Insights and possible solutions", describes the role of curcumin in targeting inflammatory mediators in Diabetes. 

The manuscript could after some necessary changes:

1. The molecular mechanisms are needed to be described in detail.

2. In the formulation section, more details are desired. Further, besides nano-formulation other novel drug delivery systems developed for curcumin and tested on diabetes models should be incorporated.

3. Just below the figure some extra text ("In summary, curcumin has been explored recently that targets inflammatory mediators in diabetes mellitus resulting in treatment of diabetes and its related complications; however, existing data is insufficient and need more exploration and clinical studies as well to conclude the overall effects and outcome."), entirely different from the legend of the figure is present. Kindly check for such undesirable mistakes in the manuscript.

4. The manuscript requires English editing for some sections.

Author Response

Reviewer: 1
1. The molecular mechanisms are needed to be described in detail.

Response: Thanks for the valuable suggestion. We have described the molecular mechanisms in more detail by adding more relevant citations to the respective section.

  1. In the formulation section, more details are desired. Further, besides nano-formulation other novel drug delivery systems developed for curcumin and tested on diabetes models should be incorporated.

Response: Thanks for this comment. We have added more useful details based on other novel drug delivery systems developed for curcumin and tested on diabetes models.

  1. Just below the figure some extra text ("In summary, curcumin has been explored recently that targets inflammatory mediators in diabetes mellitus resulting in treatment of diabetes and its related complications; however, existing data is insufficient and need more exploration and clinical studies as well to conclude the overall effects and outcome."), entirely different from the legend of the figure is present. Kindly check for such undesirable mistakes in the manuscript.

Response: We appreciate this important comment. We have addressed this issue and also checked the whole MS for such mistakes.

  1. The manuscript requires English editing for some sections.

Response: Thanks for this comment. The MS has been cheeked and edited thoroughly for English. We have removed the typo and syntax errors. The MS overall English has been improved accordingly.

Reviewer 2 Report

Authors wrote a review article: "How Curcumin Targets Inflammatory Mediators in Diabetes: Therapeutic Insights and possible solutions".

Introduction is well written and covered with proper references.

Page 5
"Curcumin is acceptable at large doses (12 g/day) in humans, according to phase I clinical studies"
A proper reference should be inserted.

Page 10. "Metabolic syndrome was induced through administration of 20% fructose and the animals were treated with curcumin and
Lactobacillus acidophilus individually as well as in combination."
Was 20% of fructose enough for a strong metabolic syndrome? Could authors comment on that?

Page 13. "and surely will lessen the burden of diabetes related complications"
Authors should not exaggerate and should soften this statement.

Author Response

Reviewer: 2

  1. "Curcumin is acceptable at large doses (12 g/day) in humans, according to phase I clinical studies"
    A proper reference should be inserted.

Response: Thanks for this comment. We have inserted proper reference to the mentioned text.

  1. Page 10. "Metabolic syndrome was induced through administration of 20% fructose and the animals were treated with curcumin and
    Lactobacillus acidophilus individually as well as in combination."
    Was 20% of fructose enough for a strong metabolic syndrome? Could authors comment on that?

Response: We appreciate this comment. This text has been re-checked and modified in order to make it more understandable for the readers.

  1. Page 13. "and surely will lessen the burden of diabetes related complications"
    Authors should not exaggerate and should soften this statement.

Response: Thanks for this valuable comment. We have revised this sentence and modified the text accordingly.

We look forward to your positive response.

Regards

Prof. Dr. Haroon Khan

Department of Pharmacy, Abdul Wali Khan University Mardan, Mardan 23200, Pakistan

Round 2

Reviewer 1 Report

I recommend acceptance of this paper